# Semi-automated tracking of iceberg B43 using Sentinel-1 SAR images via Google Earth Engine

YoungHyun Koo[1], Hongjie Xie[1], Stephen F. Ackley[1], Alberto M. Mestas-Nuñez[1], Grant J. Macdonald[1], Chang-Uk Hyun[2]

[1] Center for Advanced Measurements in Extreme Environments, University of Texas at San Antonio, San Antonio, TX 78249, USA

[2] Department of Energy and Mineral Resources Engineering, Dong-A University, Busan 49315, Republic of Korea

*Correspondence to*: Hongjie Xie (Hongjie.Xie@utsa.edu)

**Abstract.** Sentinel-1 C-band synthetic aperture radar (SAR) images can be used to observe the drift of icebergs over the Southern Ocean with around 1-3 days of temporal resolution and 10-40 m of spatial resolution. The Google Earth Engine (GEE) cloud-based platform allows processing of a large quantity of Sentinel-1 images, saving time and computational resources. In this study, we

process Sentinel-1 data via GEE to detect and track the drift of iceberg B43 during its lifespan of 3 years (2017-2020) in the Southern Ocean. First, to detect all candidate icebergs in Sentinel-1 images, we employ an object-based image segmentation (simple non-iterative clustering – SNIC) and a traditional backscatter threshold method. Next, we automatically choose and trace the location of the target iceberg by comparing the centroid distance histograms (CDHs) of all detected icebergs in subsequent days with the CDH of the reference target iceberg. Using this approach, we successfully track the iceberg B43 from the Amundsen

Sea to the Ross Sea, and examine its changes in area, speed, and direction. Three periods with sudden losses of area (i.e. split-offs) coincide with periods of low sea ice concentration, warm air temperature, and high waves. This implies that these variables may be related to mechanisms causing the split-off of the iceberg. Since the iceberg is generally surrounded by compacted sea ice, its drift correlates in part with sea ice motion and wind velocity. Given that the bulk of the iceberg is under water (~30-60 m freeboard and ~150-400 m thickness), its motion is predominantly driven by the westward-flowing Antarctic Coastal Current which

dominates the circulation of the region. Considering the complexity of modeling icebergs, there is a demand for a large iceberg database to better understand the behavior of icebergs and their interactions with surrounding environments. The semi-automated iceberg tracking based on the storage capacity and computing power of GEE can be used for this purpose.

## 1. Introduction

When a large ice mass breaks off from an ice shelf or glacier into the ocean, it forms an iceberg. An iceberg has a lifespan of several years or longer and its area ranges from a few square kilometers to thousands of square kilometers. Considering the majority of an iceberg is under water, iceberg drift is a good indicator of ocean circulation (Collares et al., 2018). Since the trajectories and speeds of icebergs also depend on multiple and complex environmental variables (e.g. ocean, atmosphere, sea ice, etc.), icebergs provide important insights for the interaction of these variables (Schodlok et al., 2006). In addition, the formation and melting of icebergs influences global climate (Romanov et al., 2008; Mackie et al., 2020), ocean flux (Silva et al., 2006; Rackow et al., 2017; Starr et al., 2021), sea ice production (Martin et al., 2007; Merino et al., 2016), dissolved iron concentration (Lin et al., 2011; De Jong et al., 2015), and ecosystems and biology (Wilson et al., 2016; Schwarz and Schodlok, 2009; Biddle et al., 2015). Furthermore, icebergs can threaten ship navigation (Lasserre, 2015)**.** Therefore, detecting and tracking icebergs is extremely important to understand the changing sea ice, ocean, and atmosphere in the polar regions.

Radar remote sensing, including scatterometer and synthetic aperture radar (SAR), is an efficient tool for monitoring both movements and area changes of icebergs. While multispectral images can be useful for observing icebergs, they cannot be used during polar night or under cloudy conditions. In contrast, SAR images can be used for analysis regardless of the weather conditions or time of year (Han et al., 2019; Mazur et al., 2017; Wesche and Dierking, 2012). In particular, although scatterometers facilitate daily position and motion observations of large icebergs with a coarse spatial resolution (Budge and Long, 2018; Stuart and Long, 2011b, a), SAR instruments have a more significant advantage in precise observations of iceberg area changes due to their relatively fine spatial resolution. Indeed, various SAR instruments have been used for detecting or tracking icebergs in the polar oceans including ERS-1 (Young et al., 1998; Willis et al., 1996), ENVISAT (Li et al., 2018; Howell et al., 2004; Mazur et al., 2017; Barbat et al., 2019, 2021), Radarsat-1 (Wesche and Dierking, 2015; Lane et al., 2002; Power et al., 2001), Radarsat-2 (Scheuchl et al., 2004; Denbina and Collins, 2014), TerraSAR-X (Frost et al., 2016), and Sentinel-1 (Lopez-Lopez et al., 2021; Moctezuma-Flores and Parmiggiani, 2017; Heiselberg, 2020; Han et al., 2019).

Most of these SAR studies used the higher backscatter contrast of icebergs to distinguish icebergs from the surrounding sea ice or water (Mazur et al., 2017). In addition, recently, instead of traditional pixel-based approach, image segmentation methods have become popular to reduce the speckle degradation of SAR images and efficiently detect icebergs (Lopez-Lopez et al., 2021; Mazur et al., 2017; Barbat et al., 2019). Although the iceberg detection is commonly conducted automatically, the tracking is still usually conducted by manually finding the images containing the target iceberg (Moctezuma-Flores and Parmiggiani, 2017; Parmiggiani et al., 2018; Li et al., 2018; Mazur et al., 2019), which requires a lot of time and labor so restricts the number of processed images. Moreover, even if both detection and tracking processes are automatized, overall process requires downloading a large number of satellite images (Barbat et al., 2021), which takes long time and needs a huge storage capacity.

In this respect, Google Earth Engine (GEE), a cloud computing platform for geospatial analysis by Google (Gorelick et al., 2017), can be a promising tool to automatize the detection and tracking of icebergs. GEE makes it possible to process a large volume of geospatial data without downloading them onto local computers. Although various satellite images are freely accessible via GEE in near real-time, Sentinel-1 is the only SAR data that has been provided by GEE as individual images in near real-time. As the constellation of the two twin satellites (Senitnel-1A/B), Sentinel-1 has a remarkable temporal (< 2-3 days in the polar regions) and spatial resolution (≤ 40 m) (Torres et al., 2012). A number of studies have taken advantage of the unprecedented computing performance of GEE for the Sentinel-1 data processing: glacier margin mapping (Lea, 2018), glacier lake mapping (Zhang et al., 2020), estimation of glacier surface speed (Di Tullio et al., 2018), crop mapping (Mandal et al., 2018; Singha et al., 2019; Jin et al., 2019), flood mapping (DeVries et al., 2020; Clement et al., 2018), and wetland mapping (Mahdianpari et al., 2019, 2020). Despite the variety of applications of Sentinel-1 data in the GEE platform, there have not yet been any published studies

that leverage GEE's large data storage capabilities and computing power for automated tracking of icebergs.

In this study, therefore, we track the drift of an iceberg in the Southern Ocean by taking advantage of the Sentinel-1 data and the GEE platform. Our target iceberg is the iceberg B43, as designated by the National Ice Center (NIC, usicecenter.gov), which calved off from the Thwaites Glacier ice shelf in the Amundsen Sea in April 2017 (Figure 1). This tabular iceberg had a relatively large size ($> 100$ km$^2$ in April 2017) and a distinctive shape compared to surrounding icebergs. As a preliminary
investigation based on manual downloading of Sentinel-1 images and visual interpretation before applying the tracking method, we found that this iceberg drifted westward along the coast from the Amundsen Sea to the Ross Sea with a relatively stable flat-topped surface and eventually broke into several pieces in the northern Ross Sea in March 2020. The aim of this study is (1) to take advantage of the GEE cloud computing platform for its potential to track the drift and decay of this iceberg from the Amundsen Sea to the Ross Sea and (2) to assess the impact of major environmental factors (e.g. ocean currents, winds, sea ice concentration,
sea ice drift, temperature) on changes in its movement and area. In addition, we examine satellite altimeter data to estimate the freeboard and thickness of this iceberg.

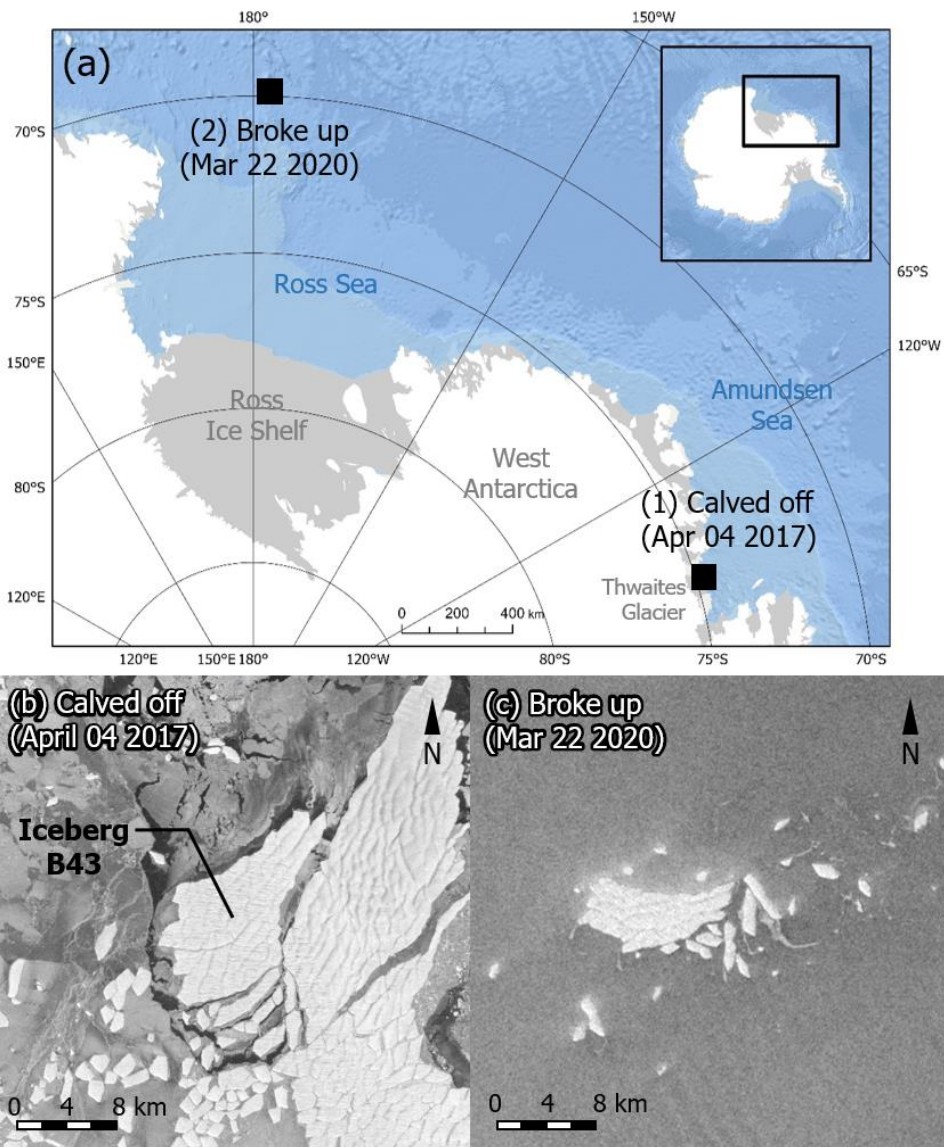

**Figure 1.** (a) Antarctic region showing the two locations where iceberg B43 calved off in the Amundsen Sea ((1) April 04, 2017)
and broke up in the Ross Sea ((2) March 22, 2020). (b) Sentinel-1 SAR image showing the ice conditions when B43 calved off at location (1) and (c) Sentinel-1 SAR image showing the ice and ocean conditions when B43 broke up, with ocean waves around at

location (2).

## 2. Data

### 2.1. Sentinel-1 SAR images

In this study, we use ESA's Sentinel-1 C-band (center frequency of 5.405 GHz, wavelength of 5.6 cm) SAR Ground Range Detected (GRD) data in order to detect the location and area of the target iceberg. This product has three different spatial resolutions of 10, 25, or 40 meters with different resolution modes, but all images are resampled to 40 m. Since each Sentinel-1 satellite (Sentinel-1A/B) has a 12-day revisit cycle, this makes the combination of Sentinel-1 A and B has a revisit cycle of 6 days; benefiting from the high latitudes of the Southern Ocean, we can acquire at least one Sentinel-1 image capturing the iceberg per 2-3 days (Torres et al., 2012). We import and process these images via the GEE Code Editor (web-based integrated development environment (IDE) for the GEE JavaScript API) and GEE Python API. Each Sentinel-1 image scene is released from ESA after being pre-processed with (1) thermal noise removal, (2) radiometric calibration, and (3) terrain correction (sentinel.esa.int). Of four available band combinations of the polarization products (VV, HH, VV+VH, and HH+HV), we use only the HH band because most of the images contain this band. In addition, we also use the angle band that represents the interpolated viewing incidence angle at each cell. During the automated tracking of the iceberg, a total of 433 images from April 2017 to March 2020 are used.

### 2.2. Satellite altimeter data

In order to estimate the freeboard and thickness of the iceberg, we use height measurements from two satellite altimeters: CryoSat-2 and ICESat-2.

CryoSat-2 employs Ku band radar (center frequency of 13.575 GHz) to measure surface heights with 400 m of along track footprint and 1.65 km of across track footprint. It operates in SAR mode over sea ice and oceanographic areas and SAR Interferometric (SIN) mode around ice sheet edges and mountain glaciers (ESA, 2019). Since the target iceberg floated in the ocean through sea ice and coastal areas, we search for both Level 2 SAR and SIN products that overlapped with the iceberg from April 2017 to March 2020.

ICESat-2's Advanced Topographic Laser Altimeter System (ATLAS) uses laser photons at 532 nm to retrieve surface heights with 11 m of laser footprint and 0.7 m of spacing (Magruder et al., 2020; Neumann et al., 2019). Trillions of photons are emitted for each pulse but the number of returned photons is around seven photons for a typical snow-covered surface (Neumann et al., 2019). ATLAS consists of six multi-beams, each with three strong beams and three weak beams, and the strong beams have four times greater energy than the weak beams (Neumann et al., 2019; Markus et al., 2017). We search for ICESat-2 ATL03 geolocated photon products (release 003) (Neumann et al., 2020) that intersected the iceberg during our study period.

### 2.3. Meteorological and sea ice motion data

We compare the drift speed, direction, and area changes of the iceberg with meteorological data from the European Centre for Medium-Range Weather Forecasts (ECMWF) ERA5 reanalysis model. We use the 2m air temperature, sea ice concentration (SIC), wind speed/direction, and wave height from the ERA5 hourly product, which has a spatial resolution of about 0.25°. This product is acquired from the Climate Data Store (cds.climate.copernicus.eu) of the Copernicus Climate Change Service.

In addition, we acquire National Snow and Ice Data Center (NSIDC) sea ice motion vector data (nsidc.org/data/NSIDC-0116) (Tschudi et al., 2019). This dataset contains daily sea ice motion vectors, which are derived from multiple data sources, including AVHRR, AMSR-E, SMMR, SSMI, and SSMI/S sensors, International Arctic Buoy Program (IABP) buoys (in the Arctic

only), and National Center for Environmental Prediction (NCEP) and National Center for Atmospheric Research (NCAR) Reanalysis forecasts. These sea ice motions are projected to the 25 km EASE (Equal-Area Scalable Earth)-Grid.

## 3. Method

### 3.1. Iceberg detection

To detect the target iceberg from the Sentinel-1 images, we employ a superpixel image segmentation method named simple non-iterative clustering (SNIC) (Achanta and Süsstrunk, 2017). This approach uses a similar concept to k-means based clustering (Simple Linear Iterative Clustering (SLIC) algorithm (Achanta et al., 2012)), but in contrast with SLIC, SNIC enforces connectivity from the start. Since GEE provides this SNIC algorithm as a basic built-in function, we use the SNIC function in the GEE environment. The object-based segmentation processes of SNIC in this study are summarized by the following steps:

(i) Apply Gaussian filter to the raw SAR image

The speckle effects of the original SAR images can introduce errors (Figure 2a). In particular, the image segmentation process can misinterpret shaded areas associated with the iceberg surface topography as artificial iceberg boundaries and split the iceberg into several pieces. To reduce this effect, we first smooth images by applying a Gaussian filter kernel within a 3-pixel radius (Figure 2b).

(ii) Initialize superpixels

In the SNIC image segmentation, all image pixels should be grouped into small clusters of connected pixels, which are named superpixels. The centroids of superpixels, referred to as seeds, are initialized with a given number of pixels chosen *a priori* on a regular grid. The size of each segment is determined by the seed value. A smaller seed value can distinigush small icebergs but takes a longer computation time (Figure 2c); a larger seed value cannot distinguish small icebergs but can reduce computation time (Figure 2d). As shown in Figure 2d, considering the size of the target iceberg, a seed value of 80 (i.e. initial centroid is spaced by 80 pixels) is adequate to detect the target iceberg B43.

(iii) Measure affinity to a centroid

The affinity of the *j*-th pixel to the *k*-th superpixel centroid is calculated by using the distance between them ($d_{j,k}$):

$$d_{j,k} = \sqrt{\frac{\|X_j - X_k\|^2}{s} + \frac{\|C_j - C_k\|^2}{m}} \qquad (1)$$

where $X_j$ is a geocoordinate (latitude and longitude) of the *j*-th pixel, $X_k$ is a geocoordinate of the *k*-th superpixel centroid, $C_j$ is the HH band backscatter of the *j*-th pixel, $C_k$ is the HH band backscatter of the *k*-th superpixel centroid, and *s* and *m* are the normalizing factors for spatial and backscatter distance, respectively. If an image has N pixels and K superpixels are expected, *s* in Equation 1 should be set to $\sqrt{N/K}$. A higher compactness factor *m* leads to more compacted superpixels and poorer boundary adherence (Achanta and Süsstrunk, 2017), and we set *m* equal to 1.

(iv) Evolution of centroid

Contrary to multiple iterations of SLIC (Achanta et al., 2012), SNIC uses a priority queue to choose the next pixel to be

160     added to a cluster. The priority queue consists of multiple candidate pixels that are 8-connected to a currently growing superpixel cluster. The pixel candidate that has the smallest distance from a centroid is selected among these pixel candidates, and the centroid value is updated online after a new pixel is added to that superpixel. This allows the SNIC algorithm to complete the updating of centroids in a single iteration with lower memory requirements. More details about the SNIC algorithm were explained in Achanta and Süsstrunk (2017).

165

        After completing the SNIC image segmentation, we detect the segments of icebergs by applying the mean brightness filter to all segments. Icebergs commonly show higher backscatter intensity compared to the surrounding sea water or sea ice (Silva and Bigg, 2005; Young et al., 1998; Mazur et al., 2017; Wesche and Dierking, 2012). Based on the bright backscatter of icebergs , we identify iceberg segments as the segments satisfying the following equation (Mazur et al., 2017):

170     $$\gamma > -0.2\,\alpha \qquad (2)$$

where $\gamma$ is the backscatter of SAR images, and $\alpha$ is the incidence angle of the SAR. When we employ this equation, we can successfully distinguish the target iceberg from the surrounding area (Figure 2e and 2f).

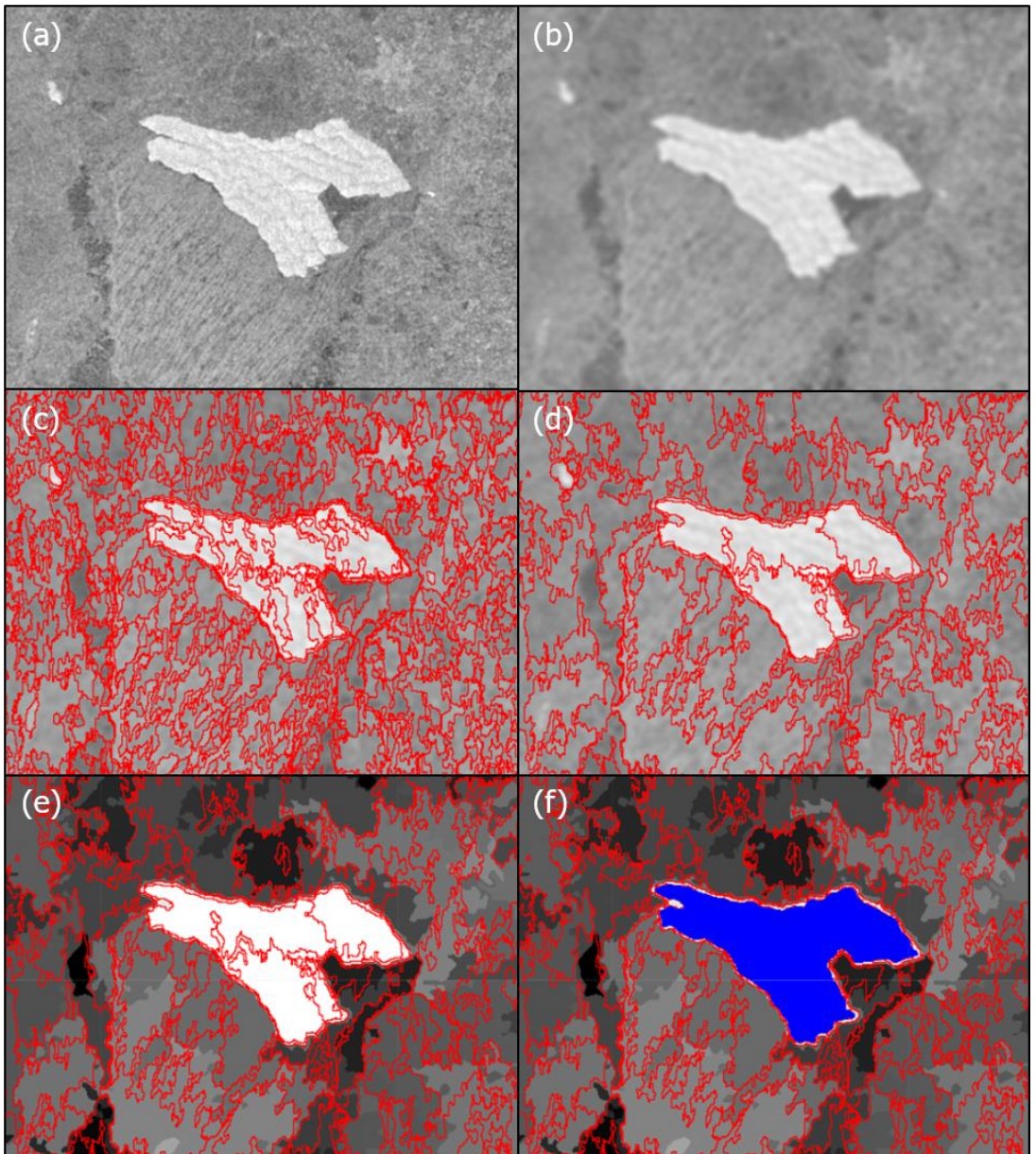

**Figure 2.** An example of the segmentation result for different levels: (a) original image (22 May 2019), (b) smoothed image by Gaussian filter, (c) Segmented image (seed = 40), (d) Segmented image (seed = 80), (e) backscatter ratio (= γ/α) image with segmented result, (f) segments identified as iceberg (blue polygon). Segmentation and other processing shown in (c)-(f) are based on the smoothed image in (b).

## 3.2. Iceberg tracking

In order to automatically detect and track only the target iceberg among multiple icebergs, we adopt a feature extraction method based on the centroid distance function (Mingqiang et al., 2008; Hasim et al., 2016; Arjun and Mirnalinee, 2015). As described in Figure 3, the semi-automated iceberg tracking process consists of the following five steps: (1) We define the target iceberg that will be tracked. We manually digitize the polygon of the reference iceberg from a Sentinel-1 image and calculate the distances from the iceberg centroid for all iceberg pixels. We make a histogram of these centroid distances for all pixels in the iceberg; this centroid distance histogram (CDH) represents the area and shape of the iceberg. (2) We search for an available Sentinel-1 image of the next day, within a 25 km radius from the reference iceberg in the current image. If the image for the next

day is not available, the search radius is extended by 25 km each day until a Sentinel-1 image is found (Figure 4a). (3) Once a Sentinel-1 image is found, all candidate icebergs are detected by the same processes as described in 3.1. (4) We calculate the CDHs for all detected icebergs. (5) By comparing the similarities of the CDHs between these icebergs and the reference iceberg, we determine which iceberg is the target iceberg for that day. The similarity of CDH between iceberg A and reference iceberg ($r(A, ref)$) is calculated by Equation 3.

$$r(A, ref) = \left(1 - \frac{\sum_{i=1}^{m}|N_A(i) - N_{ref}(i)|}{\sum_{i=1}^{m} N_{ref}(i)}\right) \times 100 \ (\%) \qquad (3)$$

where $m$ is the bin size of the histogram, $N_A(i)$ and $N_{ref}(i)$ are the number of pixels in the $i$-th bin for iceberg A and reference iceberg, respectively. We select the iceberg showing similarity greater than 80 % with the reference iceberg (Figure 4b-4e, see Appendix A). If multiple icebergs show similarity above 80 % with the reference iceberg, we select the iceberg with the highest similarity as the target iceberg. Once the target iceberg is determined, the centroid of this iceberg becomes the center of search radius for the next day in the second step. However, if the target iceberg is not detected, we return to the second step, expand the search radius, and repeat the same steps until the target iceberg is detected (Figure 3).

It is worth noting that for some days, when a significant mass of the iceberg breaks from the iceberg, the shape of the iceberg changes considerably. In these cases, the initial CDH would no longer be representative of the iceberg and a new reference iceberg would have to be manually set. Three manual reference adjustments are needed in this study, including days 18 March 2018, 25 March 2019, and 17 January 2020. In addition, if a Sentinel image is not found for more than five successive days, the search radius becomes too large to be processed in the limited memory of GEE. In that case, we must manually find the next available Sentinel image and day, and 40 such images are manually found in this study. Thus, for processing the total of 433 images, 43 manual interventions (3 manual reference adjustments + 40 manual image search) are required (i.e. ~90 % of the total processes are automated). Because of this need of occasional human intervention, we call our overall procedure semi-automated method.

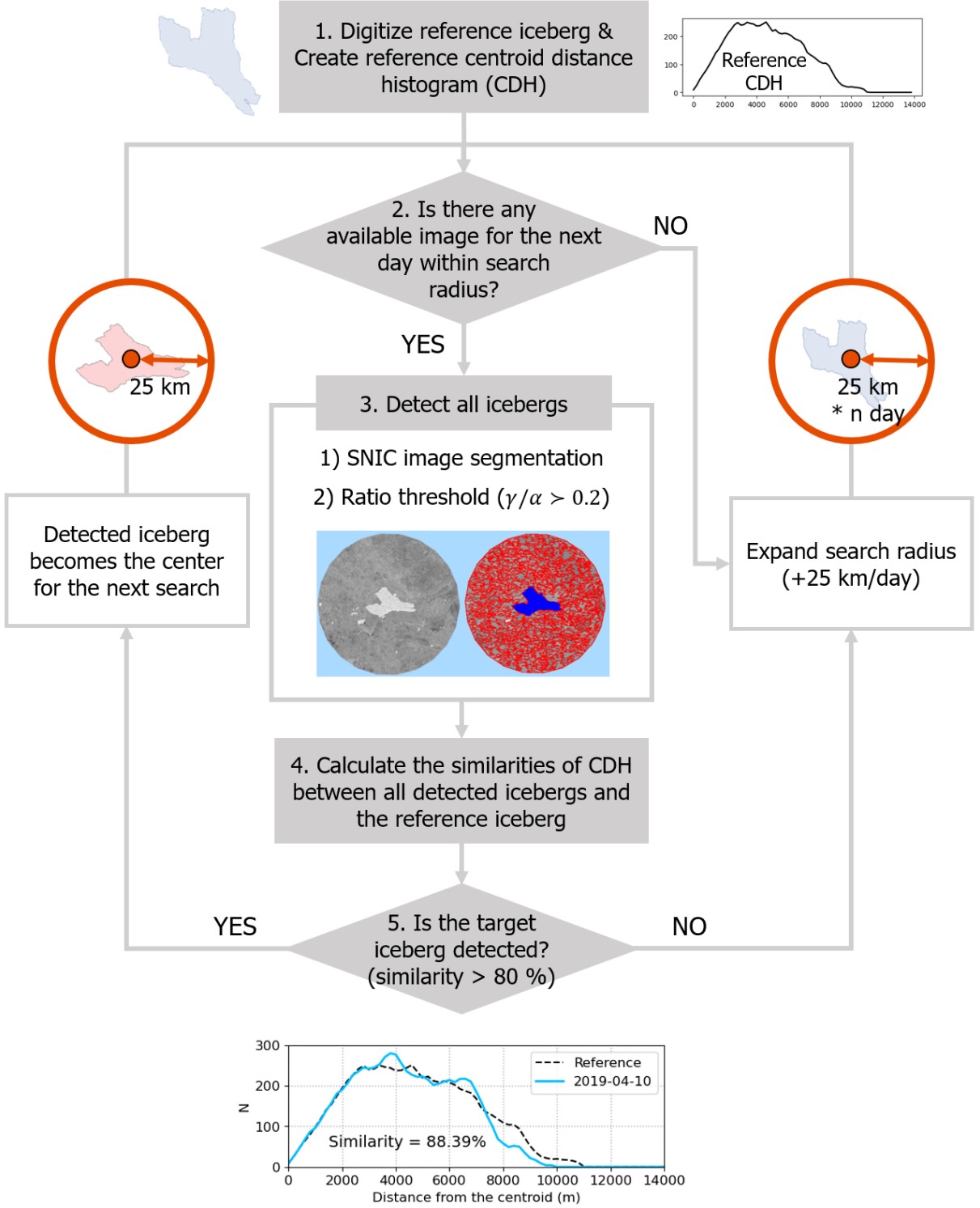

**Figure 3.** Flow chart illustrating the steps of the semi-automated procedure for iceberg detection and tracking. Details are explained in the text (section 3.2).


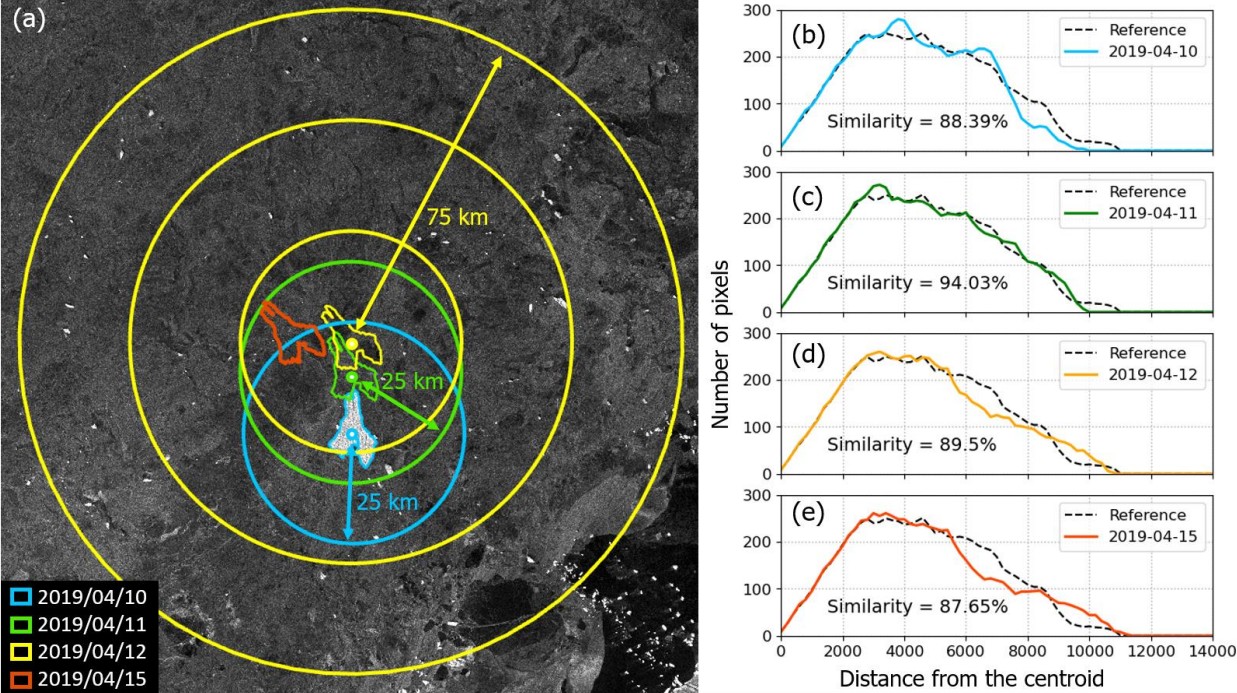


**Figure 4.** Tracking of the iceberg from 10 April 2020 to 15 April 2020. (a) Location of detected iceberg and search radius for each day, and centroid distance histograms (CDHs) of the target iceberg for (b) 10 April 2019, (c) 11 April 2019, (d) 12 April 2019, and (e) 15 April 2019.

**3.3. Calculation of iceberg thickness**

The freeboard of the iceberg (F) is defined as the difference between the heights of the iceberg body (H) and the sea surface heights ($H_{ssh}$) by the following equation:

$$F = H - H_{ssh} \quad (4)$$

For the CryoSat-2 data, we use the sea surface heights from the Level 2 SAR and SIN products, and for the ICESat-2 225 data, we use the sea surface heights from the ICESat-2 ATL10 sea ice freeboard products. We subtract these sea surface heights from the height measurements of the iceberg. Then the thickness of the iceberg (Z) can be roughly estimated from the freeboard by using Equation 5:

$$Z = \frac{\rho_w}{\rho_w - \rho_i}(F - \delta) + \delta \quad (5)$$

where $\rho_w$ and $\rho_i$ is the density of water and ice, respectively, and $\delta$ is the thickness correction of the upper firn layer. Here we 230 assume 1,025 kg/m³ and 915 kg/m³ of seawater and ice density, respectively (Griggs and Bamber, 2011; Chuter and Bamber, 2015; Zhang et al., 2020), and 15 m of typical firn layer correction (Griggs and Bamber, 2011; Lythe and Vaughan, 2001).

**4. Results and Discussion**

**4.1. Trajectory of the iceberg**

The trajectory of iceberg B43 resulted from the semi-automated iceberg tracking is shown with solid colored circles in Figure 5. The trajectory agrees well with that from the NIC iceberg archive which is reproduced with a black dashed line in Figure 5. In contrast to our semi-automated tracking procedure, the NIC trajectory is manually retrieved by ice analysts every week using a combination of SAR, visible, and infrared remote sensing images (usicecenter.gov). As shown in Figure 5, B43 drifted from the

Amundsen Sea in April 2017 into and through the Ross Sea in March 2020. The trajectory shows that B43 initially drifted mainly toward the north and northwest until about April 2018, then towards the west until November 2018, then towards the southwest until June 2019, and finally towards the northwest until it broke off. This trajectory agrees with the modeled trajectories of icebergs over the Amundsen Sea to the Ross Sea area (Gladstone et al., 2001; Merino et al., 2016; Rackow et al., 2017).

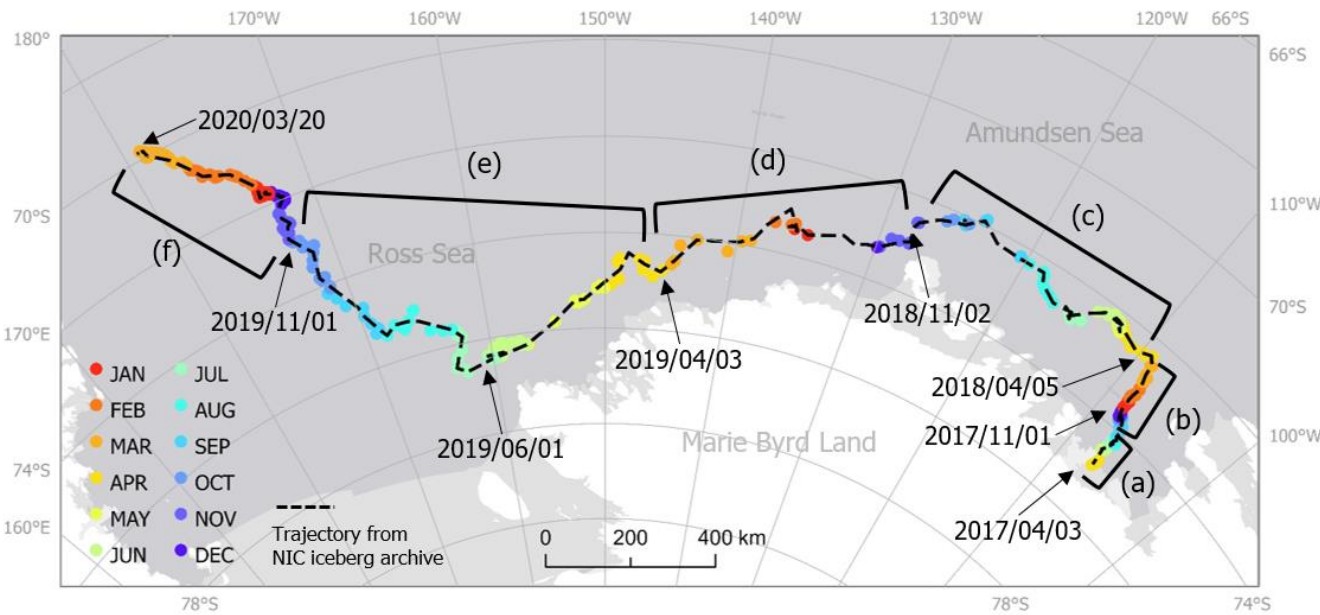


**Figure 5.** The trajectory of iceberg B43 from our semi-automated algorithm (solid colored circles) and that obtained from the NIC iceberg archive (black dashed line). The circle colors indicate the calendar month. The six square brackets labeled (a) to (f) indicate track segments selected to investigate associations with ice motions and winds.

Figure 6 shows rose diagrams for iceberg velocity along with fields of average sea ice concentration, average ice motion, and average wind velocity averaged for the six selected periods shown in Figure 5. Initially the iceberg moved relatively slowly. From April 2017 to March 2018, a speed < 4 km/day was detected more than 90% of the time and similarly a speed < 2 km/day was detected more than 75% of the time. The primary drift direction in this period was toward the west or northwest (50 % of the time for April 2017-October 2017 and 60 % of the time for November 2017-March 2018), but occasionally it also drifted eastward

or northeastward (40 % of the time for April 2017-October 2017 and 25 % of the time for November 2017-March 2018). During this period, iceberg B43 was mostly surrounded by slow and compacted sea ice (Figure 6a and 6b).

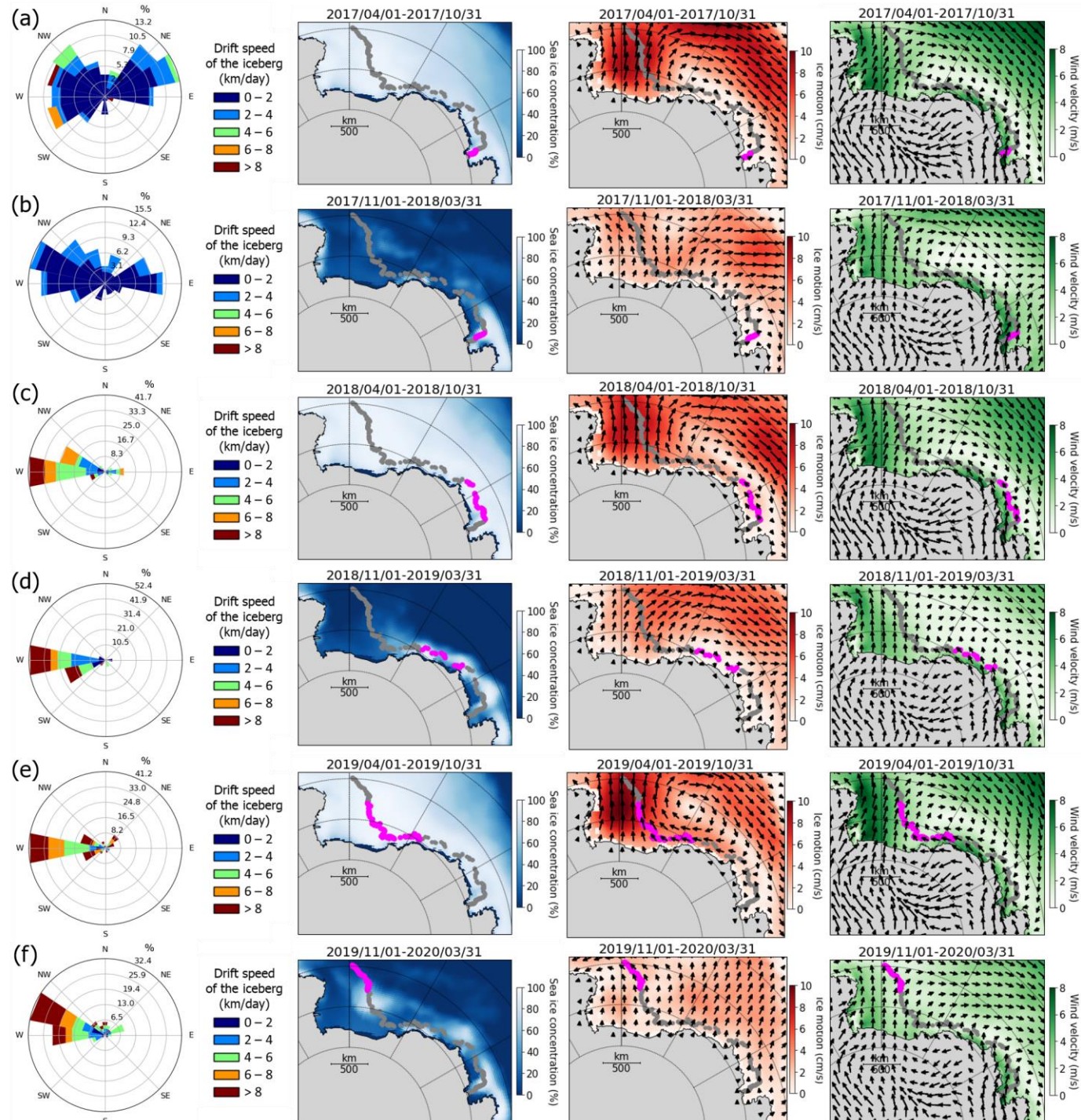

**Figure 6.** Multipanel plots with left-to-right columns respectively indicating rose diagrams of iceberg velocity, average sea ice concentration (SIC), average ice motion velocity, and average wind vector velocity. The rows correspond to the six averaging periods selected in Figure 5: (a) April 2017-October 2017, (b) November 2017-March 2018, (c) April 2018-October 2018, (d) November 2018-March 2019, (e) April 2019-October 2019, and (f) November 2019-March 2020. The entire track of the iceberg is indicated in all the maps (grey solid circles) as well as the track segments for each of the six periods (magenta solid circles).

As B43 moved away from the starting point, however, its speed increased. From April 2018 to May 2019, the iceberg mainly moved westward parallel to the coastline more than 90 % of the time, which is consistent with the westward sea ice drift and wind velocity (Figure 6c-6e). During this journey from the Amundsen Sea to the eastern Ross Sea, the drift speed was faster than 4 km/day approximately 70 % of the time, and even greater than 8 km/day 10-40 % of the time. This increase of drift speed

can mainly be attributed to the faster sea ice movement in this region compared to the starting point in the Amundsen Sea.

When B43 reached the Ross Sea (June 2019), it started to drift more in a northwestward direction (Figure 6e and 6f). From April 2019 to October 2019 the iceberg drifted at its fastest speed as it passed through an area with the fastest sea ice motion (Figure 6e). It is also noted that the directional change from westward to northwestward during this period agrees with the direction of sea ice movement and wind direction in this region (Figure 6e). It is generally known that sea ice starts to move faster from this region because sea ice is thinner in the Ross Sea compared to the Amundsen Sea, so sea ice is more affected by wind forcing (DeLiberty et al., 2011). Although the drift speed slightly decreased after November 2019, the iceberg was still observed to drift at > 8 km/day around 30% of the time for the last five months (Figure 6f). During the last five months, sea ice may have had less impact on the drift of B43 because it was surrounded by thinner sea ice (DeLiberty et al., 2011) or open water.

Figure 7 compares the drift speed and the direction of the iceberg with SIC, wind velocity, and sea ice motion. Here we merely focus on the qualitative comparison of the drift direction and speed with other climatological data. Considering that this iceberg moved along with highly compacted sea ice during most of its journey (SIC > 80 %, Figure 7a), the movement of the iceberg is likely to be related to the sea ice drift and wind velocity (Lichey and Hellmer H., 2001; Schodlok et al., 2006). Wind velocity and ice motion generally show similar variations for each other, and they both show substantial correlation with the iceberg drift (Figure 7b-7e). However, the impact of wind or sea ice drift on the iceberg drift could vary depending on the season, location, SIC, and ocean current.

Although we do not directly analyze ocean current data here due to the lack of reliability and availability of ocean current data under high-concentration sea ice, ocean current would account for a significant part of the iceberg drift because about 80-90 % of the iceberg is under the sea surface. Hence, the westward drift of this iceberg mainly represents the sea current along the Amundsen Sea to the Ross Sea. In the coastal regions of the Amundsen Sea and the Ross Sea, the ocean circulation is dominated by the westward Antarctic Coastal Current (ACoC) which is mainly driven by wind stress and buoyancy (Whitworth III et al., 1985; Orsi et al., 1995; Mathiot et al., 2011; Kim et al., 2016; Stern et al., 2016).

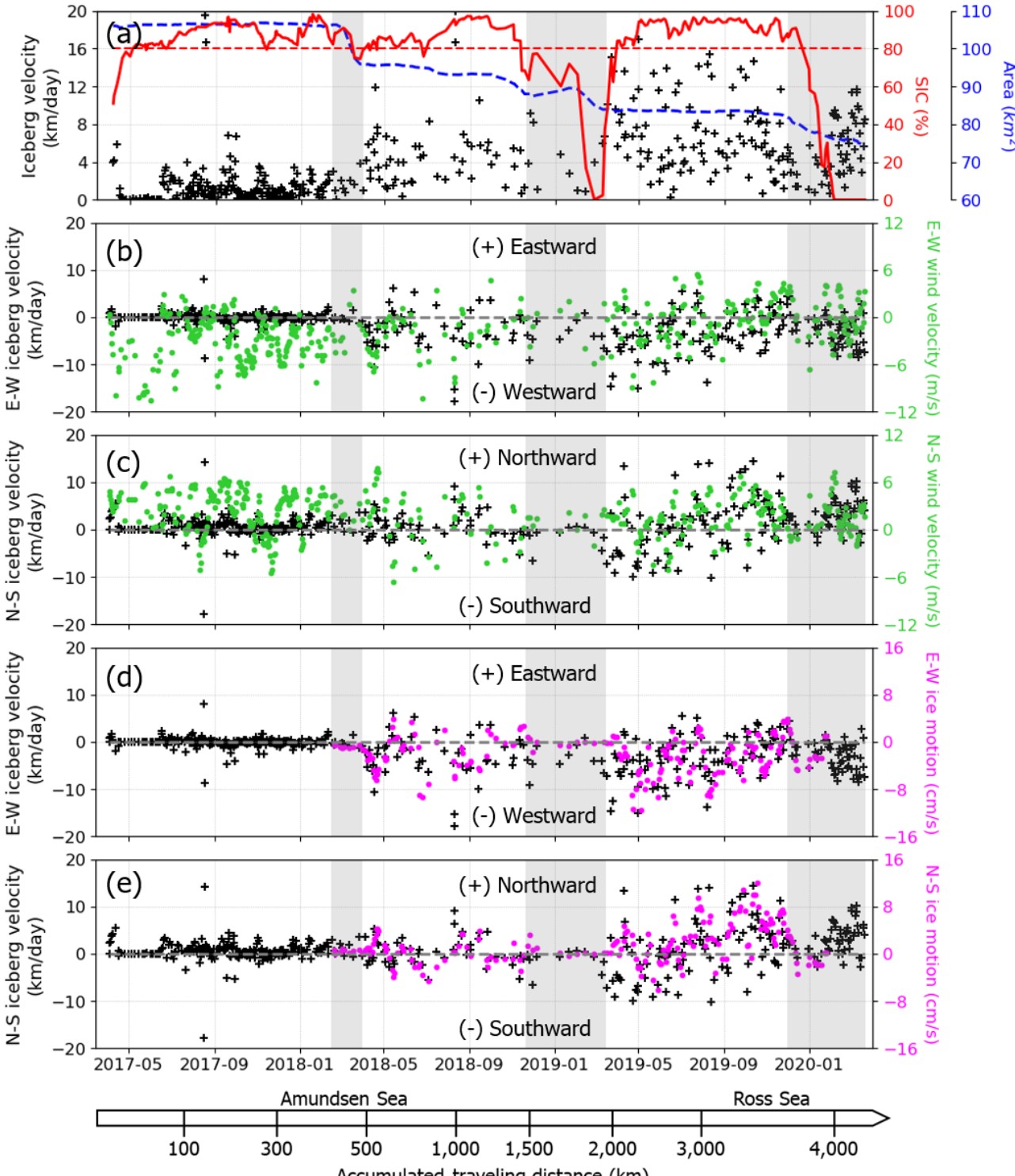

**Figure 7.** (a) Drift speed of the iceberg (black crosses), SIC (red solid line), and area changes (blue dashed line); (b) east-west velocity of both the iceberg (black crosses) and the wind (green circles); (c) north-south velocity of both the iceberg (black crosses) and the wind (green circles); (d) east-west velocity of both the iceberg (black crosses) and sea ice motion (magenta circles); (e) north-south velocity of both the iceberg (black crosses) and sea ice motion (magenta circles). Gray shaded areas indicate the split-off events of the iceberg.

## 4.2. Area variations of the iceberg

The iceberg area is also retrieved by using our semi-automated iceberg tracking method (Figure 8a). We find three noteworthy periods, all in summers or end-of-summers, when large reductions of the iceberg area are observed: (1) February-March 2018, (2) December 2018-March 2019, and (3) December 2019-March 2020 (gray shaded areas in Figure 8). We observe that a significant portion of the iceberg split away from the main iceberg during these periods. Except for these summer months, the area of the iceberg remained relatively constant or decreases gradually during the journey. Initially, from April 2017 to February 2018, the iceberg area remained at a nearly constant value of 105-107 km$^2$. The area then decreased rapidly to 95 km$^2$ around February-March 2018. From then onwards the iceberg's area decreased gradually, falling under 90 km$^2$ around December 2018 to March 2019. The iceberg area then remained nearly constant at about 84 km$^2$ during April-November 2019 until it started to decrease again and eventually broke down into several smaller pieces in December 2019-March 2020.

Although our iceberg tracking method extracts a good monthly mean trend of the iceberg area, there are several anomalous estimates of the individual area at a daily level (Figure 8a). One cause for this is due to the nearby small icebergs. A number of small icebergs around the coastal Amundsen Sea region are sometimes adjacent to the iceberg B43 (Appendix A). In this case, our iceberg-detecting algorithm recognizes them as a single iceberg body, which leads to overestimation of the iceberg area. Second, although we assume a high radar backscatter from the iceberg, this backscatter can vary by sensor incidence angle, iceberg surface, and sometimes weather conditions (Wesche and Dierking, 2012). The variations in backscatter of the iceberg can cause uncertainties in area estimation. In addition, if sea ice has a high backscatter, the similar backscatter between sea ice and the iceberg makes it difficult to distinguish the exact boundary between them (Mazur et al., 2017). Finally, we need to consider the spatial uncertainties introduced when SAR images are projected into the map coordinates and are resampled into 40 m resolution.

We compare the area changes with the ancillary meteorological data (Figure 8b-8d). It is noted that three split-up events occurred only in lower SIC areas and during higher air temperatures (Figure 8b and 8c). In higher SIC areas, the iceberg remains relatively stable because it is surrounded by sea ice (Stuart and Long, 2011b). However, when SIC decreases and the iceberg is exposed to open water, the interaction between waves and the iceberg can trigger the breakup of the iceberg (England et al., 2020; Scambos et al., 2008; Wagner et al., 2014). Given that this breakup mechanism is somewhat related to the water temperature and waves (Wagner et al., 2014), we deduce that SIC and temperature can have substantial impacts on the breakup events. In particular, the iceberg was completely broken into several small pieces in March 2020, after being continually exposed to open water since December 2019. There is a possibility that the rising wave heights during this period accelerate the breakup event (Figure 8d) (MacAyeal et al., 2006). Nevertheless, considering the complexity of the breakup mechanism, it is challenging to clearly determine which factor most contributed to the entire breakup of the iceberg B43 in March 2020.

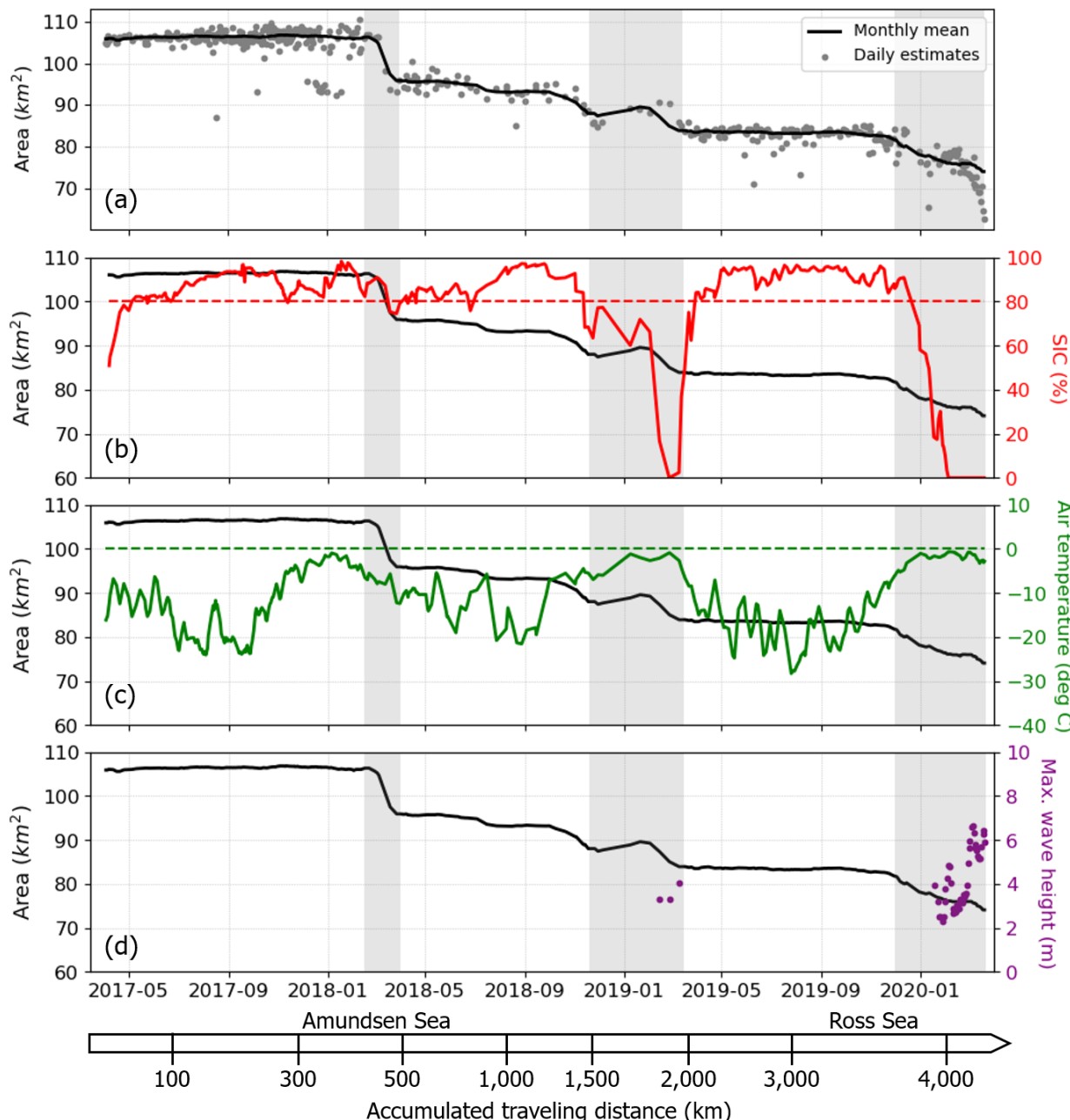

**Figure 8.** (a) Temporal changes of the iceberg area from April 2017 to March 2020. Comparison of area changes of the iceberg (black line) with (b) sea ice concentration (SIC) (red), (c) air temperature (green), and (d) maximum wave height (purple) from ERA5. Wave heights are not calculated for unmarked periods because the high SIC effectively dampens waves. Gray shaded areas indicate the split-off events of the iceberg.

### 4.3. Freeboard and thickness of the iceberg

To estimate the freeboard and thickness of the iceberg, we search for coincident CryoSat-2 and ICESat-2 tracks overlapping with the iceberg. We find one overlapping CryoSat-2 track and four overlapping ICESat-2 tracks that can be used to estimate the freeboard of the iceberg (Table 1). Although there are time differences between the Sentinel-1 images and the altimeter

measurements, we conclude that these CryoSat-2/ICESat-2 tracks include the heights of the iceberg by comparing the drift of the iceberg and the height measurements. For example, Figure 9 shows the Sentinel-1 image on 31 March 2019, and the CryoSat-2 track on the same day. Although the higher points of CryoSat-2 do not exactly correspond to the iceberg in the Sentinel-1 image

345 due to the time differences between the two datasets (around 14 hours), we can assume that these higher points represent the surface of the iceberg when the movement of the iceberg is considered. In particular, in the Sentinel-1 image, we cannot see any potential 40 m-height objects except the iceberg. Furthermore, near the iceberg, the CryoSat-2 points are off the straight ground track line, which indicates that there is an object different from the surrounding sea ice or open water. Since the retracking algorithm of the CryoSat-2 SIN product may emphasize a late-returned signal (Wingham et al., 2006), the radar signal from the edge of the iceberg

350 can be biased to the non-iceberg area. Therefore, we assume that the heights represent the surface heights for the iceberg, and the mean freeboard is 42.4 m for this date (Table 1).

  While CryoSat-2 has only one coincident track with the iceberg, there are four coincident ICESat-2 tracks passing through the iceberg (Table 1). Figure 10 shows an example of the ICESat-2 ATL03 data on 29 November 2019, and the Sentinel-1 image on the next day. Although they have time differences of about 24 hours, the heights of the ATL03 tracks are likely to represent the

355 heights of the iceberg at the time of the Sentinel-1 image. It is interesting that ICESat-2 data describe the surface height of the iceberg in more detail with some ridges and valleys (Figure 10). This is attributed to a better spatial resolution of the ICESat-2 ATL03 product (11 m of footprint spaced by 0.7 m in the along-track direction (Magruder et al., 2020)) than the CryoSat-2 SAR or SIN product (~400 m of resolution in the along-track direction). In addition, the multiple tracks of ICESat-2 can be a considerable advantage for examining the shape of the iceberg surface.

360  As shown in Table 1, the iceberg freeboard is estimated to be greater than 50 m in November 2018, but less than 50 m from then on. However, the limited number of height measurements is not sufficient to help us to find a significant trend of freeboard change. Moreover, since each track passed through a different part of the iceberg, the heights also depend on the sampling points where CryoSat-2 or ICESat-2 track were passing by. Nevertheless, we can safely deduce that the freeboard of this iceberg was within 30-60 m during the tracking period. This range of freeboard agrees with the previous estimates for other large icebergs

365 over the Antarctic (Tournadre et al., 2015; Scambos et al., 2005; Romanov et al., 2012; Han et al., 2019). As shown in Table 1, this level of freeboard is equivalent to the iceberg thickness of 150-400 m (Equation 5). This implies that about 120-350 m of the iceberg thickness (i.e. 80-90 % fraction of the total thickness) is below the water surface. Thus, considering the average thickness (~265 m), initial area (~105 km$^2$), and 15 m firn layer of this iceberg, the melting of the entire iceberg body will contribute to approximately 24 Gt of freshwater input into the Southern Ocean. This amount of freshwater is equivalent to about 1 % of total

370 annual freshwater flux in the Southern Ocean (Hammond and Jones, 2016).

**Table 1.** Freeboard and thickness of the iceberg estimated by ICESat-2 (IS2) and CryoSat-2 (CS2)

| Date | Data | Latitude | Longitude | Freeboard (m) | Thickness (m) |
|---|---|---|---|---|---|
| 2018-11-13 | IS2 ATL03 | -72.93 | -127.58 | 52.59 ± 8.33 | 365 ± 78 |
| 2018-12-24 | IS2 ATL03 | -73.22 | -131.70 | 37.21 ± 6.55 | 222 ± 61 |
| 2019-03-31 | CS2 SIN | -74.62 | -144.98 | 42.40 ± 5.88 | 270 ± 55 |
| 2019-11-29 | IS2 ATL03 | -72.12 | -172.64 | 34.70 ± 8.90 | 199 ± 83 |
| 2020-02-07 | IS2 ATL03 | -71.13 | -174.87 | 42.19 ± 3.86 | 268 ± 36 |

375

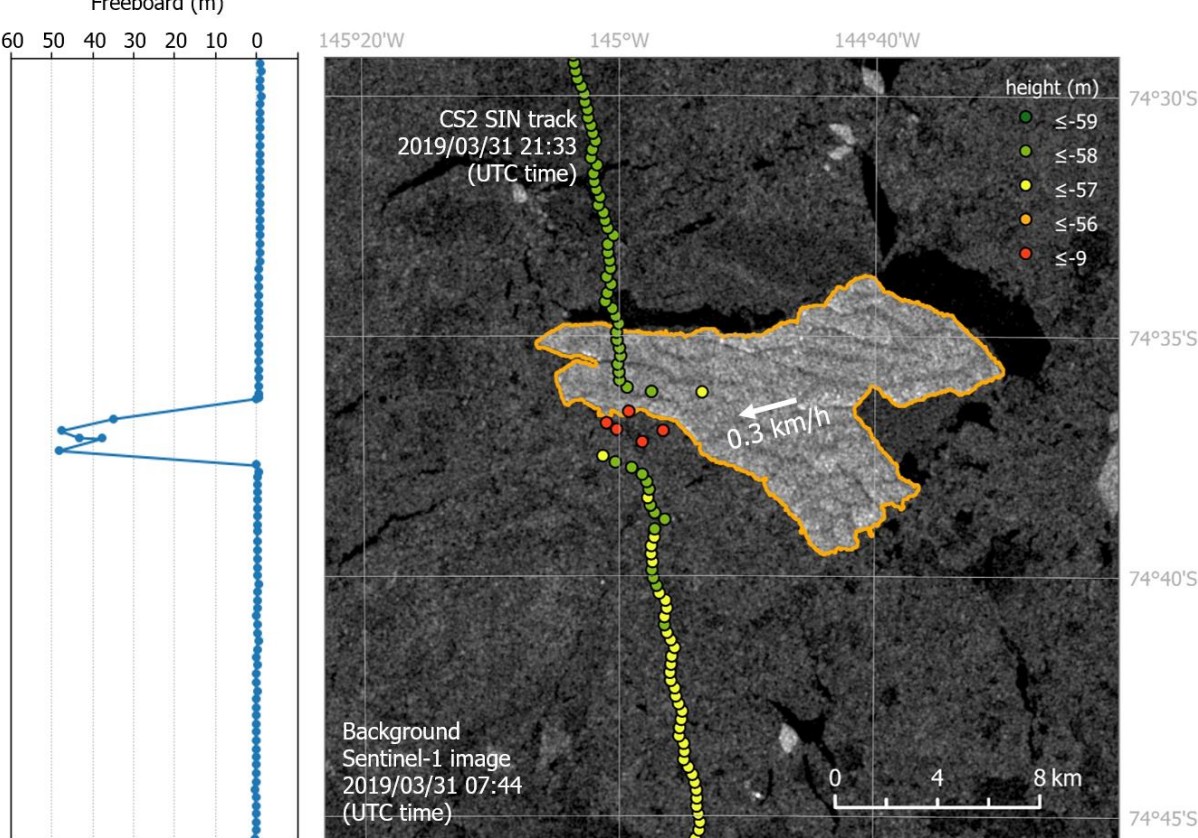

**Figure 9.** Freeboard measures from CS2 SIN track on 31 March 2019, and Sentinel-1 image 14 hours before. Considering the drift of the iceberg between the image and the CS2 SIN track, the highest points (red dots) are the intercepted points on the iceberg by the CS2 SIN track.

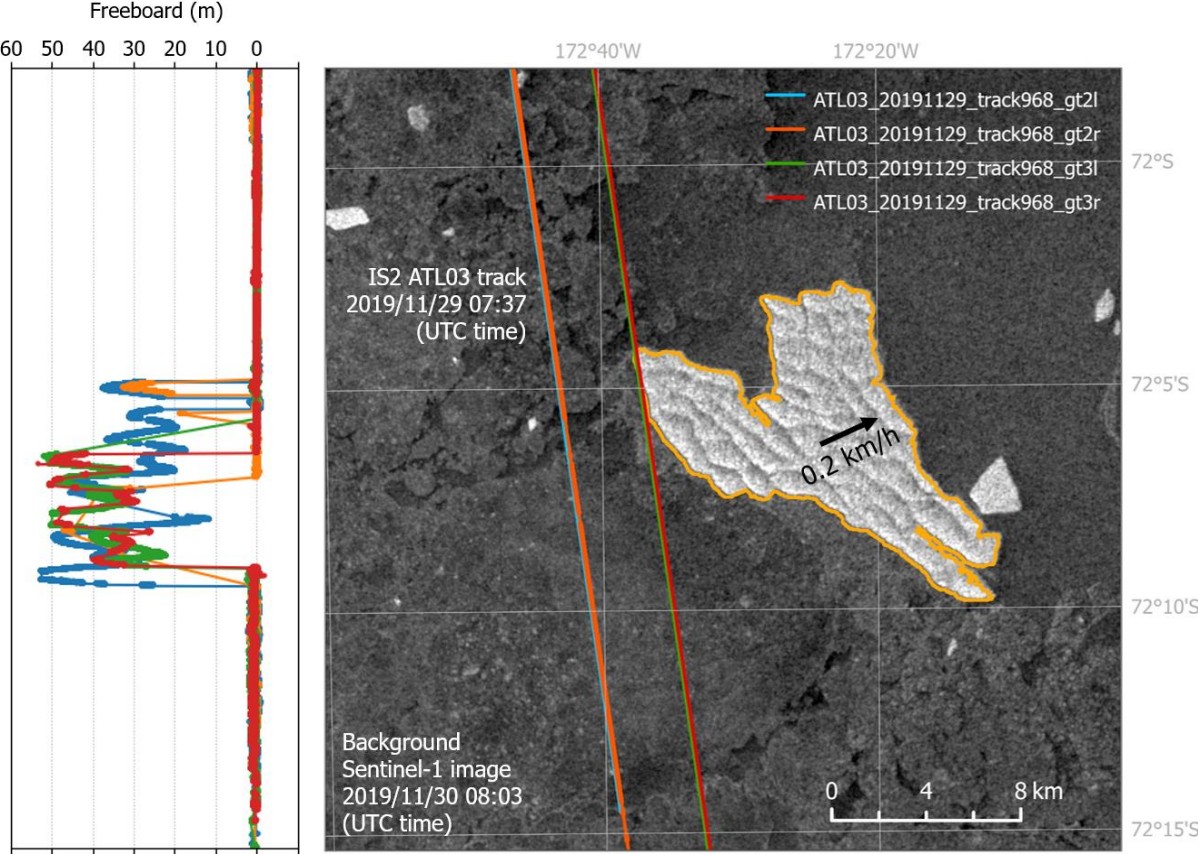

**Figure 10.** Freeboard measurements from ICESat-2 ATL03 tracks on 29 November 2019, and Sentinel-1 image on 30 November 2019 (about 24 hours later).

## 5. Conclusions

This study demonstrates for the first time the potential of Google Earth Engine (GEE) for iceberg tracking in the Southern Ocean using Sentienl-1 SAR images. It presents a cloud-based computational method for tracking iceberg B43 from when it calved off the Amundsen Sea coast until it broke up in the Ross Sea. First, iceberg B43 is detected by using a SNIC object-based image segmentation and backscatter threshold algorithm. Then, the trajectory of this iceberg is constructed by comparing its centroid distance histogram with those of icebergs in subsequent images. Although manual tracking is used for several dates due to memory limitations of GEE and occasional large changes in the iceberg shape, the method (with a ~90% automation) is successful in tracking B43 for three years from April 2017 to March 2020.

Factors affecting the decrease in the area of iceberg B43 over time are studied by comparison with contemporaneous changes in environmental variables from ERA5 reanalysis data and NSIDC sea ice motion data. First, in terms of SIC, we find that split-up events of the iceberg occurred with lower SIC and higher exposure to open water conditions. This suggests that wave action, as calculated in ERA 5, played a role in triggering the break-up of the iceberg. Our analysis suggests that higher water temperature and higher wave heights can also accelerate this break-up mechanism. In addition, we find that iceberg drift could be associated with sea ice drift or wind velocity. The speed and direction of the iceberg drift agrees with the variations of sea ice drift and winds in the Amundsen Sea and Ross Sea regions. Given that this iceberg was 150-350 m in thickness, with the majority of its volume underwater, the main contribution to the westward drift of the iceberg is thought to be the westward sea current (ACoC) from the coastal Amundsen Sea to the Ross Sea.

Predicting the behavior of icebergs (e.g. trajectory, speed, decay, and breakup) is a highly complex endeavor that depends on many factors related to dynamic and thermodynamic interactions between ocean currents, the atmosphere, waves, sea ice, and bottom topography (Stern et al., 2016; Rackow et al., 2017; Schodlok et al., 2006; Lichey and Hellmer H., 2001; Gladstone et al., 2001). Thus, developing, validating, and calibrating models for predicting iceberg's variables would require a large and accurate iceberg database containing the life history of many icebergs. The method presented here can be used for these purposes by taking advantage of GEE's large data storage capability and high computing power. Indeed, GEE allows us to process 433 Sentinel-1 SAR images in the cloud, taking only 10-30 seconds to process and analyze one day of data and only a few hours to process all three years. Moreover, since all image-processing tools are provided by GEE, this approach is able to save time, cost, and resources as compared to a similar traditional tracking done in a local computer. Therefore, iceberg tracking based on Sentinel-1 images and the GEE platform would be an efficient option for tracking a large number of icebergs in the polar oceans.

To build a large database of icebergs using the GEE-based iceberg tracking approach presented in this paper, however, three key limitations would need to be overcome. First, in terms of the image segmentation, although the SNIC algorithm in GEE detects iceberg B43 successfully (Figure 2), this superpixel approach should be tuned for a further application to small icebergs. In this study, the surface roughness and shape of iceberg B43 have no significant impacts on the image segmentation because the iceberg is large and superpixels are also large (i.e., seed = 80). However, for detecting small icebergs, the surface shapes of icebergs can have significant impacts on the image segmentation and iceberg detection because the superpixels should be small. Therefore, further enhanced image segmentation process may be required for the detection of small icebergs. Second, although the centroid distance histogram (CDH) based tracking approach successfully works for iceberg B43 thanks to the distinctive shape and large size of the B43, this method has limitations in tracking small icebergs that share similar shapes or areas. Indeed, we attempted unsuccessfully to track the small pieces of B43 when it broke up (Figure 1c) because these pieces have similar CDHs. On the other hand, we also note that our tracking method may have difficulty tracking icebergs that are too large to be captured within a single Sentinel-1 scene. Third, our tracking algorithm is not fully-automated. In this study, instead of defining the detected iceberg from the previous day as the reference for the next-day tracking, we defined the reference iceberg at the initial digitizing step and use this reference for all the following steps. Although this approach enables us to avoid tracking errors caused by temporary anomalies (e.g. impacts of surrounding small icebergs), it prevents our algorithm from detecting sudden changes in the iceberg shape or area. By overcoming these limitations, fully-automated tracking of small icebergs would be possible and a large and accurate iceberg database would be constructed in the future.

## Appendix A. Variations of similarity values

In this study, we track the target iceberg that has > 80 % of similarity with the reference iceberg. This appendix is for discussing how these similarity scores change and what factors potentially affect the similarity scores. Figure A1 shows the area of iceberg B43 and the similarity of CDH with the reference iceberg. 80 % threshold of CDH similarity is appropriate for tracking this iceberg, but the similarity decreased near the split-off events. Hence, if these split-off events reduce the similarity to < 80 %, it is necessary to newly define the reference iceberg. Figure A2 shows the Sentinel-1 images for six low-similarity dates in Figure A1(b). In Figure A2, we observe three factors that are likely to lower the CDH similarity between the target iceberg and reference iceberg: (1) surrounding small icebergs (A, B, C, E in Figure A2), (2) split-off events (F in Figure A2), and (c) uncertainty from the image segmentation because of confusion between shaded part of the iceberg and surrounding bright sea ice (B, D in Figure A2).

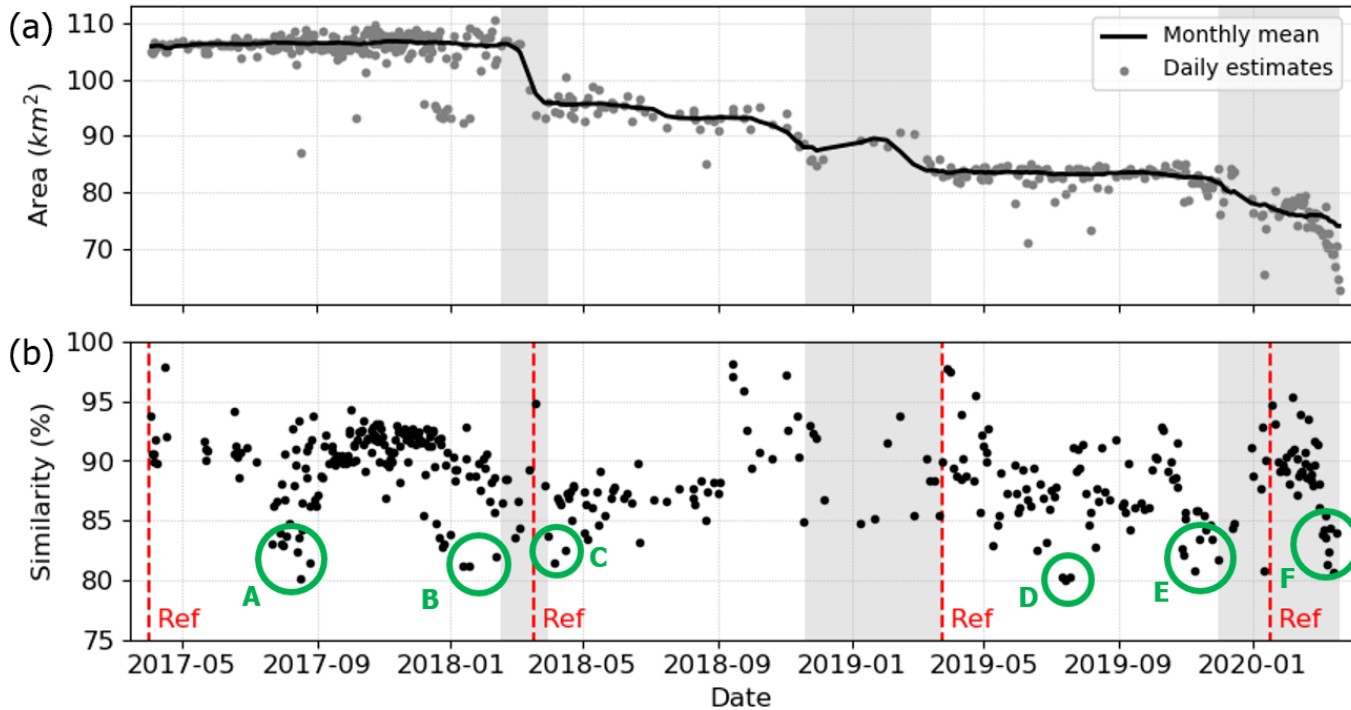


**Figure A1.** (a) Area changes of the tracked iceberg B43 (same to Figure 8a). Gray shaded areas indicate the split-off events of the iceberg. (b) Variations of similarity values by time. Red vertical dashed lines indicate the timing of manual digitization to define the reference iceberg. Green circles A-F indicate six distinctive low-similarity dates (< 85 % similarity).

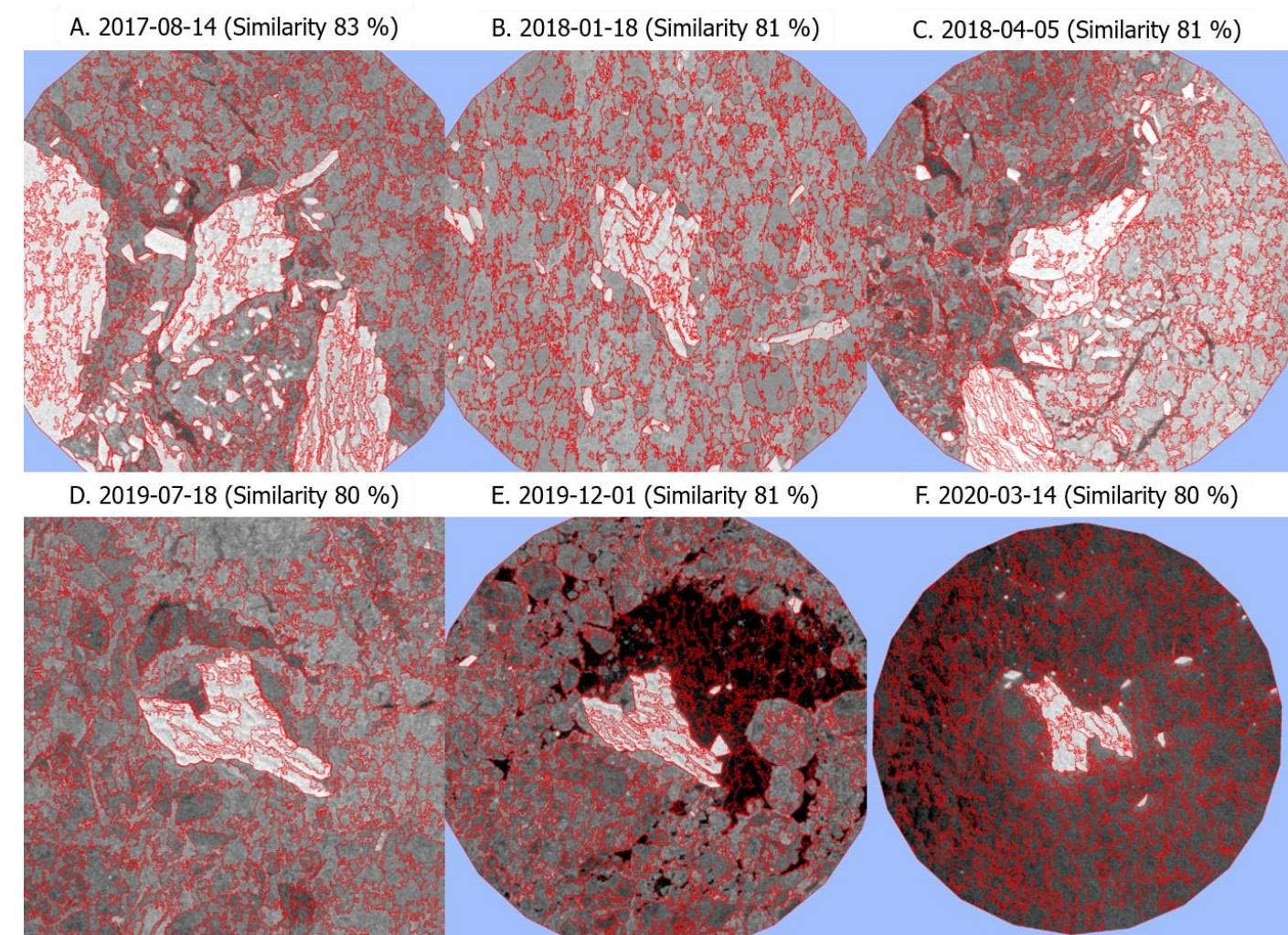

A. 2017-08-14 (Similarity 83 %)  B. 2018-01-18 (Similarity 81 %)  C. 2018-04-05 (Similarity 81 %)

D. 2019-07-18 (Similarity 80 %)  E. 2019-12-01 (Similarity 81 %)  F. 2020-03-14 (Similarity 80 %)


**Figure A2.** Sentinel-1 SAR images and SNIC segmentation results for six low-similarity times A-F in Figure A1.

*Code and data availability.* Python code for this GEE-based semi-automated iceberg tracking is available at:
https://github.com/YoungHyunKoo/GEE_iceberg_tracking.git.

*Author contributions.* YK designed the research, conducted programming and analysis, created figures, and drafted the initial manuscript. HX, SA, AM, GM, CH contributed to writing and editing of the manuscript.

*Competing interests.* The authors declare that they have no conflict of interest.

*Acknowledgments.* We would like to thank the European Space Agency (ESA) for processing and providing Sentinel-1 data and GEE for proving the cloud computation capability. Additional data: ECMWF, CryoSat-2, and ICESat-2 data from ESA and NASA are acknowledged here. Critical reviews and constructional comments from two anonymous reviewers and handling editor Dr. Stef
Lhermitte to improve the quality of this manuscript are greatly appreciated.

*Financial support.* This study is funded by NSF (1835784) and NASA (80NSSC18K0843 and 80NSSC19M0194) grants.

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
