# Peer review of "Semi-automated tracking of iceberg B43 using Sentinel-1 SAR images via Google Earth Engine"

_The Cryosphere, 2021_

## Author Response (AR1)

**Editor**

Dear YoungHyun Koo and colleagues,

Your TCD manuscript "Semi-automated tracking of iceberg B43 using Sentinel-1 SAR images via Google Earth Engine" received two constructive reviews where both reviewers identified some issues and questions, which you already clarified in your author comments. Both reviewers scored your manuscript ranging from fair to good where both suggest a revision. In this revised version I suggest you also take into account my comments made during the access-review (repeated here below) .

After your author's response, I would like to ask you now to re-upload the:

1 revised version of the manuscript

2. author's response as you have already (partlt) done in the interactive discussion, but now make it more detailed.

3. track change version so the reviewers can easily check what has been changed.

Based on this response and your revised version, I plan to reconsult the reviewers before taking a final decision.

Best regards,

Stef Lhermitte

Editor comments:

- the introduction provides a good context (i.e. GEE opportunity for iceberg tracking), but fails to address any of the topics the paper actually discusses. I miss some background on the iceberg tracking methodologies and/or the scientific questions related to the role of environmental variables on iceberg trajectories. The introduction only sketches the GEE opportunity, but fails to provide the scientific background on the real scientific questions etc.

- We add more explanations about the iceberg tracking background (L53-60). The previous methods usually use higher backscatter of icebergs, and recently image segmentation has become common for image processing of SAR data. However, most of these previous studies are based on "local" data processing, which requires large data storage space and time to process a large number of satellite images. Therefore, we focus on using the new cloud-computing platform GEE for tracking an iceberg, which does not require downloading a large number of satellite images into the local computer anymore.

- We add scientific questions about the role of environmental variables on iceberg trajectories (L33-41).

- L65-67 is a bit strange in the context of an introduction as it already contains the main result in the introduction

- Before performing "automatic tracking" of the target iceberg, we need to briefly investigate basic information of this iceberg. This part is conducted as a preliminary study about this iceberg and we made it clear in the revision (L89-91).

- When introducing a new methodology (in this case the GEE tracking algorithm), it is always important to benchmark it to current state-of-the-art methodologies. You compare it to NIC data, but I think this

comparison should be extended (and quantified) to show better the validity of the method. I would also start the results with the comparison with existing tracking methods and not introduce it halfway the results.

- Thank you and we agree with you. In this study, we would like to show the potential of GEE as a future "platform" to perform iceberg tracking. As we mentioned, the advantage of this GEE platform is that users do not need to download individual images so users can save their time and labor compared to the traditional local data processing (L53-71, L460-465). Thus, we need to prove that our tracking shows a reliable result compared to the existing iceberg database. As far as we know, the NIC iceberg archive is the only data source that contains the trajectory of the iceberg B43. They did it by manually checking various satellite images weekly and determine the weekly location of the iceberg, which needs a lot of labor and experienced individuals. Thus, we compare our iceberg trajectory with the NIC iceberg archive to show the reliability of our tracking result.

- However, here we only conduct "qualitative" comparison for the following reason: while NIC iceberg archive provides very rough information about the iceberg (e.g. weekly-scale movement of the iceberg, km-scale width and length), our iceberg tracking is more detailed (e.g. daily-scale movement of iceberg, m-scale iceberg size). Hence, we just qualitatively check if our iceberg trajectory agrees with the NIC iceberg archive, rather than the quantified validation.

- We move the iceberg trajectory (4.1) prior to the iceberg area (4.2).

- I helps readability if results and discussion sections are separated.

- At first, we divided the result and discussion sections. However, for this specific study, we determined to combine the result and discussion sections because it is more efficient and more readable to explain the results and reasons together (e.g. area of iceberg and the reason for the area change, trajectory of iceberg and the reason for the direction/speed changes). It was somehow difficult for us to clearly separate result and discussion for this case. We rather keep it as it is. We hope you can agree.

- I would recommend you to change the color palette of Fig 7 (right column). Currently, this figure uses rainbow palettes. There are many good reasons to avoid rainbow palettes (see https://www.nature.com/articles/s41467-020-19160-7?s=09), so I think it is important to communicate with more correct color palettes.

- Thank you. Done. (See new Figure 6)

- As the method is applied on a small iceberg, I miss some discussion on how the method would work on larger icebergs (e.g. A68) that often are not captured within a single S1 scene.

- We add a mention about very large icebergs. (L476-477)

**RC1**

This paper describes an approach to detect, locate, and measure Antarctic tabular icebergs. With its finer spatial resolution, the data can supplement optical tracking by the U.S. National Ice Center and scatterometer tracking by the Scatterometer Climate Record Pathfinder (www.scp.byu.edu). The latter maintains an extensive database of daily iceberg positions going back to the late 1970's (www.scp.byu.edu/iceberg; Budge and Long, 2017; Stuart and Long, 2011; Long et al. 2002).

The authors should be sure to note in the second sentence of the introduction that the paper exclusively considers (large) Antarctic tabular icebergs to avoid confusion with smaller icebergs that occur in both polar regions. While the paper mainly consider SAR observations, radar scatterometers have also been extensively used for detecting and tracking icebergs. Though the scatterometers observations have much lower resolution and less precision in measuring iceberg area, the wide swath of the scatterometers facilitate daily position observations (Budge and Long, 2011; Stuart and Long, 2011). This fine temporal resolution is helpful during period of rapid iceberg motion, for example when icebergs such as A68 move from the Weddell Sea into the South Atlantic. The rapid motion and spinning of this icebeg and others such as B10A can be revealed in animations of the daily scatterometer images, e.g., www.scp.byu.edu/iceberg/A68tracking.html The authors are encouraged to include references to scatterometry as another tool for iceberg tracking.

- We really appreciate your great comments and suggestions. We agree that scatterometer is another good tool to track large icebergs. We add more descriptions and references about scatterometers in the second paragraph of introduction. (L42-48)

It has been previously noted (Budge and Long, 2017; Stuart and Long, 2011), that the disintegration of icebergs is slowed when the iceberg is encapulated with sea ice, suggesting that the encapulating sea ice may protects the iceberg from exposure to wave stress. This scatterometer-derived resul supports the authors' postulation.

- We add the references you suggested in the section 4.2, with the description that sea ice may protect icebergs from exposure to wave stress (L374-375).

References

J.S. Budge and D.G. Long, "A Comprehensive Database for Antarctic Iceberg Tracking Using Scatterometer Data," IEEE Journal of Selected Topics in Applied Earth Observations, Vol. 11, No. 2, pp. 434-442, doi:10.1109/JSTARS.2017.2784186, 2017.

K.M. Stuart and D.G. Long, "Iceberg Size and Orientation Estimation using SeaWinds", Cold Regions Science and Technology, doi:10.1016/j.coldregions.2011.07.006, Vol. 69, pp. 39-51, 2011.

K.M. Stuart and D.G. Long, "Tracking large tabular icebergs using the SeaWinds Ku-band microwave scatterometer", Deep-Sea Research Part II, doi:10.1016/j.dsr2.2010.11.004, Vol. 58, pp. 1285-1300, 2011.

D.G. Long, J. Ballantyne, and C. Bertoia, "Is the Number of Icebergs Really Increasing?" EOS, Transactions of the American Geophysical Union, Vol. 83, No. 42, pp 469 & 474, 15 Oct. 2002.

**#RC2**

In this paper, the authors do present and apply a semi-automatic approach for iceberg motion tracking from SAR imagery using the Google Earth Engine (GEE) as a base environment for data acquisition and processing. Then, is discussed the main drift patterns and area decay observed along with the B-43 iceberg correlating it with the major forces involved in the iceberg lifetime. A significant problem is addressed and, while the manuscript does not add that much new information to improve the search for reliable methods able to improve iceberg detection/tracking in the complex polar environment and also, to enhance our understanding of icebergs and their drift patterns, the paper contributes to icebergs studies bringing to light the potential of the application of the cloud-based GEE as a platform to improve massive data processing. However, the manuscript could be strengthened by putting more effort into a validation analysis about the robustness of the method (segmentation, classification e tracking scores) for different scenes in which icebergs appear with different (i) radiometric signatures, (ii) shapes, (iii) surrounded by sea-ice/ice mélange, (iv) proximity to clusters of small icebergs/ice shelves, and others issues that may be present in different SAR scenes. It could do assist the reader in the reproducibility of the method on a large scale. Once, automated approaches able to detect and tracking icebergs on large scale can be very useful to construct reliable iceberg coupled models for climate studies. Following some specific questions, that could help to make clear if the method is semiautomatic for iceberg detection or B-43 detection. As described in the manuscript, the method was applied only on the iceberg B-43 (target of the study), leading the reader to ponder about the specificity and limitations of the method that could do not allow its reproducibility for a bigger number of icebergs.

- Thank the reviewer very much for the thorough evaluation of the study, weakness and advantage of the method. We have included these into the revision, especially to the conclusion and appendix. Some of the responses are included below to your specific comments.

- The chosen iceberg is very well defined with a shape very descriptive. In the conclusions, this issue is mentioned however would be useful to elaborate better on this kind of o issue once this can be a serious limitation. Also about the need for the polygon to be manually digitized to be used as a reference to track.

-  We add more discussions about the issue of iceberg shape (L466-482). In addition, we add more details about the need of manual digitization (L200-201, L477-481, Appendix A).

- Don't you think the superpixel approach used is sensitive to the iceberg surface complexity (i.e. texture)? Although a smooth process is performed which can help to search the target B-43. Aspects such as edges definition can be decreased for smaller bergs reducing segmentation performance, maybe demanding specific segmentation tunning for all different icebergs;

- Yes. The image segmentation method is sensitive to the iceberg surface or texture. Although this was not a critical issue in detecting our target iceberg B43, this will be a critical issue for detecting other small icebergs. We add more details about this discussion for a further improvement of our method (L467-472).

- As discussed in previous papers, the classification approach based directly on the iceberg backscattering strength is more suitable for bright icebergs or icebergs in open water. This can reduce even more the automatic aspect demanding more human intervention to constantly manually find the references used to computed the centroid distance histograms (CDH's).

- Yes. It is well-known that icebergs have bright backscatter compared to the surrounding open water and sea ice. Actually we use this bright backscatter feature to identify icebergs, and CDHs are calculated from these identified icebergs. We make this clearer in the updated manuscript (L184-189), and add discussions about why the manual reference process is needed (Appendix A).

- As described, to perform motion tracking the method is sensitive to shape changes, which can increase the total number of human interventions for continuing the tracking (What is unfeasible for a large

number of icebergs). In this sense, would be very useful an analyze how much shape change the tracking method is reliable to overpass in order to keep tracking without manual intervention.

- Thank you for your great suggestions. In Appendix A, we add the analysis of similarity scores and display several Sentinel-1 images for low similarity scores. This part will help understand the weakness of our method and how to improve our method in the future.

- Furthermore, the manuscript is well-written with good-quality figures. Just a few additional analyses about the method's robustness and some comparison/discussion with other state-of-art methods could strengthen the paper.

- Thanks. We hope the revised version achieves the goal.